# The Design and Development of Instrumented Toys for the Assessment of Infant Cognitive Flexibility

**DOI:** 10.3390/s23052709

**Published:** 2023-03-01

**Authors:** Vishal Ramanathan, Mohammad Zaidi Ariffin, Guo Dong Goh, Guo Liang Goh, Mohammad Adhimas Rikat, Xing Xi Tan, Wai Yee Yeong, Juan-Pablo Ortega, Victoria Leong, Domenico Campolo

**Affiliations:** 1Robotics Research Center, School of Mechanical and Aerospace Engineering, Nanyang Technological University, Singapore 639798, Singapore; 2Singapore Centre for 3D Printing, School of Mechanical and Aerospace Engineering, Nanyang Technological University, Singapore 639798, Singapore; 3School of Mechanical and Aerospace Engineering, Nanyang Technological University, Singapore 639798, Singapore; 4Division of Psychology, School of Social Sciences, Nanyang Technological University, Singapore 639798, Singapore; 5Division of Mathematical Sciences, School of Physical and Mathematical Sciences, Nanyang Technological University, Singapore 637371, Singapore; 6Department of Pediatrics, University of Cambridge, Cambridge CB2 1TN, UK

**Keywords:** instrumented toys, ecological behavioural assessment, executive function development, inertial motion detection, barometric force sensing, 3D printing

## Abstract

The first years of an infant’s life represent a sensitive period for neurodevelopment where one can see the emergence of nascent forms of executive function (EF), which are required to support complex cognition. Few tests exist for measuring EF during infancy, and the available tests require painstaking manual coding of infant behaviour. In modern clinical and research practice, human coders collect data on EF performance by manually labelling video recordings of infant behaviour during toy or social interaction. Besides being extremely time-consuming, video annotation is known to be rater-dependent and subjective. To address these issues, starting from existing cognitive flexibility research protocols, we developed a set of instrumented toys to serve as a new type of task instrumentation and data collection tool suitable for infant use. A commercially available device comprising a barometer and an inertial measurement unit (IMU) embedded in a 3D-printed lattice structure was used to detect when and how the infant interacts with the toy. The data collected using the instrumented toys provided a rich dataset that described the sequence of toy interaction and individual toy interaction patterns, from which EF-relevant aspects of infant cognition can be inferred. Such a tool could provide an objective, reliable, and scalable method of collecting early developmental data in socially interactive contexts.

## 1. Introduction

Executive functions (EFs) are higher-order cognitive control mechanisms that are commonly conceptualised as a triad of mental skills comprising inhibitory control, working memory, and cognitive flexibility [1]. These mental abilities support complex thinking skills, such as reasoning and creative problem-solving [2], and influence the development of socioemotional competencies, such as the Theory of Mind [3]. They are considered essential for mental and physical health [4,5,6].

Cognitive flexibility refers to switching between tasks, rules or dimensions and adapting one’s behaviour to a changing environment [7,8]. Categorisation tasks test an infant’s ability to flexibly categorise objects based on different dimensional features (or attentional sets), such as shape versus material [9,10]. An infant’s mental categorisation of objects can be inferred from a behavioural measure of sequential touching. If infants sequentially touch objects from the same category (e.g., balls) more often than expected by chance, it is inferred that they are doing so because they perceive these objects to belong to the same category [11,12,13]. Horst and colleagues (2009) [14] found that 14–18-month-old infants could flexibly adapt their categorisation of objects by either a perceptually salient dimension of taxonomic distinction (e.g., shape) or a less salient dimension (e.g., deformability). An example study protocol for this task is available at this URL: https://psyarxiv.com/dejns.

Changes in the pattern of an infant’s object touch sequences can index mental set-shifting, i.e., shifts in the mental dimensional set that infants use for categorisation, such as shape or compressibility. One common performance measure is mean run length (MRL), where run lengths are the number of touches in a row to objects from the same category (i.e., shape). MRLs are calculated by dividing the total number of touches by the total number of runs across all categories [15]. The calculated MRLs are then compared against a Monte Carlo simulation’s average “random” sequence lengths to assess performance against chance [14]. In addition to these classic measures, newer indices based on the conditional probability of infant touch sequences may provide further insight into the cognitive strategies adopted by infants during this task.

However, to detect sequences of object touches made by infants, these events must first be manually extracted and coded from video footage by a trained human experimenter. Furthermore, it is possible to measure the effect of maternal scaffolding on infant mental set-shifting via the introduction of a brief period of maternal social interaction during which the mother demonstrates object compressibility to the infant. The quantification of maternal behaviour during this period of social interaction similarly relies on the hand-coding of relevant behaviour.

Accordingly, this paper presents the design and validation of a set of instrumented toys as an objective, reliable, and scalable form of cognitive flexibility task instrumentation and a behaviour measurement tool that can complement and accelerate manual means of data coding (Figure 1). It can extract typically coded measures, such as when a toy is touched, how long the infant interacts with it, and the overall touching sequence. We can additionally measure squeezing patterns that the infants exhibit to validate mental set-shifting from shape-based to material-based categorisation. Such instrumented toys may, in the future, inform the development of scalable quantitative measurement tools and screening methods for the assessment of the early development of executive function skills in young children.

## 2. Functional, Technical, and Physical Specifications

### 2.1. Requirements and Existing Setup

To evaluate mental set-shifting in infants during the object categorisation task, the number and sequence of touches of each object by the infant need to be identified. The current methodology relies on human coders manually extracting these touch sequences by watching video recordings of infants playing with toys. The set of eight toys comprises two hard balls (79 g), two soft balls (23 g), two hard cubes (80 g), and two soft cubes (5 g) (Figure 2). All the cubes have sides measuring 50 mm, and all the spheres have diameters measuring 55 mm and are all different colours. A latent factor currently not measured, which could be useful in reinforcing the identification of mental set-shifting in infants based on the less salient dimension of deformability, would be the squeezing patterns of the infant’s grasp on the toys.

### 2.2. Detecting Toy Interaction

To scale up the object categorisation task for lab-based and ecological environments, we need to automatically detect when and how an infant interacts with the toys. Motion tracking enables the desired automatic extraction of interaction and motion patterns. This can be implemented using various technological solutions, as shown in [16]; however, not all solutions are suitable for use in ecological environments. We also need to keep costs low and ensure the system is easy to set up and manage. Based on technological assessments to identify suitable techniques for motion tracking, optical and inertial sensing seem to be the most favourable and widely used [17,18].

Optical markerless methods can be used on recorded videos that are part of the existing object categorisation task paradigm. Combining human pose detection, object detection, and multi-object tracking algorithms, key points can be detected and tracked throughout the video (Figure 2) [18,19,20]. However, the accuracy of the underlying detection algorithms determines the performance, which suffers from the issue of occlusions. It also requires a structured environment restricted to the area within the camera’s field of view.

Alternatively, inertial measurement unit (IMU)-based motion and orientation tracking eliminate line of sight problems and structured environment requirements, making it more appealing from an ecological perspective. These methods are a compact, low-cost, and robust way to detect the motion and orientation of objects. They have been extensively used in lab-based and ecological studies of infant motor development [21,22,23] and are the ideal method to detect motion and interaction for our application.

### 2.3. Detecting Toy Squeezing

To detect the squeezing of the toys, we need to detect the forces applied to them using force, pressure, or tactile sensors. Several different working principles for such sensors were explored by [24,25,26] and have been used in previous infant-related research. A sensorised ball designed by Campolo et al. [27] used quantum tunnelling composites (QTC), which change its electrical resistance based on changes in applied force [28] in order to detect grasping patterns during manipulation. Cecchi et al. [29] incorporated piezoresistive pressure sensors and flexible force sensing resistor (FSR) sensors in sensorised toys to measure infants’ reaching and grasping. Serio et al. [30] used pressure sensors connected to air chambers to measure the amplitude of the force applied for quantitative monitoring and measuring infants’ motor development.

Tenzer and Jentfot [31,32] developed a versatile, low-cost, and sensitive tactile sensor using commercial off-the-shelf MEMS barometers, and they commercialized it as TakkTile [33]. It has been used in robotics by Ades et al. [34] and Koiva et al. [35] to sense grasping events using robotic grippers. The working principle of the MEMS barometer-based tactile sensor is the communication of surface contact pressure within a layer of rubber to the ventilation hole of the sensor and, thus, to the MEMS transducer. Similarly, Takada et al. [36] and Quinn et al. [37] have used a waterproof mobile device’s built-in barometer to measure touch force, which works on a similar principle. When an airtight or waterproof device is touched, the distorted surface changes the air pressure inside that device and, thus, changes the built-in barometer value [36,37]. This ability to detect forces through changes in internal pressure makes barometric tactile sensing suit our requirements for a low-cost, versatile, and sensitive way to detect squeezing forces on the toys.

### 2.4. Proposed Platform

For the object categorisation task, balls and cubes help assess an infant’s cognitive flexibility. The selected sensors need to be integrated into a platform that can be used with minimal alterations to existing paradigms. We proposed using instrumented toys to measure the development of EF in infants in a scalable manner in lab-based and ecological environments.

Campolo et al. [27] designed a sensorised ball to analyse the development of perceptual and motor skills in ecological environments. IMUs have been embedded in toys to assess spatial cognition [21,22,38] and detect possible autism spectrum disorders (ASD) at an early stage [23]. Pressure and force sensors were used to study infants’ grasping actions [29,30,39]. A whole suite of instrumented toys was developed to provide early intervention for infants at risk for neurodevelopmental disorders and reduce parental stress [40,41].

These examples demonstrate the viability of instrumented toys for integrating sensors to assess infant development. However, instrumented toys targeted explicitly at measuring the development of EF in infants are yet to be developed, and a need exists for such tools.

## 3. Instrumented Toy Design and Fabrication

### 3.1. Sensor Core

Using commercially available sensors, particularly in infant behavioural research tool development, has the advantage of being certificated for public use while ensuring high quality and safety standards are met. These certifications reduce the potential risk of harm when using instrumented toys that contain sensors and batteries in particular. It allows us to leverage existing expertise in sensor development while focusing on designing, developing, and deploying instrumented toys at scale without compromising accuracy.

We used Physilog 6^®^ (P6), a commercial off-the-shelf sixth-generation wearable motion sensor platform produced by MindMaze Assessments (Figure 3). It comprises an inertial measurement unit (IMU) with nine degrees of freedom (DoF) and a barometer that is typically used for human motion and gait analysis [42,43]. Its versatility allowed us to capture, measure, and analyse the necessary parameters for detecting the touching and squeezing of the toys during the object categorisation task.

Each instrumented toy had a P6 sensor embedded into it to detect the touching and squeezing of the toys. The specifications of the P6 sensor are summarised in the user manual document provided by the manufacturer [44]. The IMU and barometer were configured to sample at 64 Hz for easy data synchronisation and maximising battery life while striking a suitable temporal resolution. The P6 sensor has an internal clock that can be synchronised to a PC’s clock, which allows for easy synchronisation with other sensors, opening up the possibility of multi-modal data analysis for studying EF development. Multiple P6 sensors can communicate and synchronise with each other using the built-in BLE communication protocol. One sensor acts as the master, wirelessly broadcasting the clock signal on a particular channel, and the others act as the client listening to this signal. Data stored onboard as “.BIN” files are sorted into timestamped folders accessed by connecting the sensor to a PC via the USB-C interface and downloading it.

Dragon Skin^™^ 20, a high-performance platinum-cured liquid silicone compound, was used to seal the hole of the P6 sensor’s barometer to use it as a tactile sensor. The silicone will transduce the squeezing forces on the surface to changes in internal pressure that the barometer can detect. A custom 3D-printed mould (Figure 4a) was used to enclose the sensor and shape the silicone layer surrounding it. The silicone was cured overnight (Figure 4b,c) and unmoulded to create the final sensor core for the instrumented toys (Figure 4d). The manufacturer, Smooth-On, Inc., Macungie, PA, USA, provides a detailed material safety data sheet and skin-safe certification, making Dragon Skin^™^ 20 a suitable nontoxic silicone material for use in our application where infants might come in direct contact with the material [45,46].

### 3.2. Physical Structure

The physical structure of the instrumented toy encloses the sensors, creates the final shape, and controls the rigidity. The weight and dimensions of the physical structures of the instrumented toys were defined by the anthropometry of an infant’s hand, the existing regular toys used, and the physical dimensions of the sensor (44 mm × 36 mm × 17 mm) to be embedded. The cube has a length of 60 mm, and the sphere has a diameter of 65 mm.

Modern 3D printing technologies and advances in material science have enabled the fabrication of complex structures using materials of varying rigidity [47,48,49,50]. Ansys SpaceClaim’s Faceted Shell and Infill tool was used to design an octahedral lattice that minimises mass, ensures uniform stiffness, and avoids needing a support structure during fabrication. The lattice structure was designed with a wall thickness of 1 mm, a unit cell size of 10 mm, and a strut thickness of 1 mm, which translates to 11% infill density (Figure 5).

Fused deposition modelling (FDM) 3D printing using thermoplastic polyurethane (TPU) filament was used to fabricate the octahedral lattice physical structure (Figure 6). The hard toys were printed using a 95A shore hardness TPU filament, while the soft toys were printed using an 85A shore hardness TPU filament. The print parameters used to print the lattice structure were as shown in Table 1. The 3D-printed TPU parts are inert and nontoxic, making them suitable for use in our application where infants will come into direct contact with the material [51,52].

### 3.3. Integration

The instrumented toys were designed using a modular multilayered approach to decouple the sensing capabilities from the physical properties. The sensing core determines the modality of the data that can be captured, while the physical structure determines the real and perceived affordances of the toy by the infant. The risk of harm and injury to the infant is further minimised by placing the sensors and battery at the toy’s core. The physical structure, fabricated using nontoxic and noncombustible materials, acts as a barrier. This instrumented toy design paradigm can be expanded to utilise other sensors within the sensing core that can be enclosed by different physical structures based on the desired play and interaction style. For the set of instrumented toys to be used in the object categorisation task to assess cognitive flexibility in infants, the sensor core records motion and pressure data from which we infer the toys’ sequence of touching and squeezing. The 3D-printed lattice structure defines the toy’s affordance by varying the ball and cubes’ colour, size, and rigidity.

## 4. Squeezing Detection

### 4.1. Experimental Setup

A preliminary quantitative experiment to verify if the barometer of the P6 sensor embedded in each toy could detect squeezing was performed using a Kinova Gen3 7 DoF robotic arm (Figure 7) to repeatedly and consistently squeeze each toy ten times. An ATI Industrial Automation Net Force/Torque Sensor Mini40 measured the reaction forces ($F_{sensed}$). A 3D-printed holder was fastened to the robot end effector and force sensor to hold the toys in place and control the toy’s contact area. The cubes had a contact area of 72.00 cm${}^{2}$, and the balls had a contact area of 64.84 cm${}^{2}$ between the robot end effector and the force sensor. It was noted that studies on infant grip force within the first 12 months had measured the range to be between 5 to 35 kPa [29,30,39,40,53]. As such, to validate performance in a minimal force application condition, the robot was programmed in position control mode to move 10 mm vertically from $z_{0}$ to $z_{1}$ with a velocity of 65 mm/s to apply a force $F_{robot}$ = 20 N on the toys. The expected pressure applied on the cubes was 2.77 kPa and on the balls it was 3.08 kPa, well under the typical grip strength range. Before beginning the experiment, each object was placed on the force sensor and its weight was zeroed out. Data from the robot and the force sensor were timestamped and logged to a PC. The internal time of the P6 sensor was synchronised with the PC clock.

### 4.2. Results and Discussion

The robot consistently moved 10 mm to squeeze the toys 10 times with a regular force of $F_{robot}$ (Figure 8). The compliance of the instrumented toy’s structure produced a difference between the force applied by the robot and the force measured by the force sensor (Table 2). The soft cube absorbed 12%, the hard cube absorbed 10%, the soft ball absorbed 18%, and the hard ball absorbed 14% of the applied force. Based on the force applied by the robot and the area of contact of the 3D-printed holders, we estimated the pressure applied to each toy when squeezing. As expected, the pressure exerted ranges from 2.77 to 3.39 kPa. This did not saturate the barometer as its specified dynamic range is 100 kPa (26 to 260 kPa).

In Table 2, we can see the summary of the barometer readings across 10 squeezes for all the different toys. Noise in the barometer signal is smoothed out using a 5 Hz low-pass filter. The squeezing force applied to the toys produced a change in internal pressure that was successfully recorded by the barometer (Figure 8). The soft toys recorded a smaller change in pressure than the hard ones due to the compliance of the physical structure absorbing some of the force applied.

A drift in the baseline pressure reading of the barometer in the soft cube (Figure 8c) was observed. A drift of approximately ±20 Pa was present across all sensors either due to air leaking or getting trapped within the 3D-printed structure or the sensor core through the USB-C connector port. This drift was not of particular concern as the sensor could still consistently pick up on the dynamic changes in pressure caused by the actual squeezing of the toy. However, applying a 0.5 Hz high-pass filter helped filter out such a drift and the baseline offset of the ambient room pressure as well (Figure 9).

Furthermore, our barometric squeezing detection validation was based on 10% of a typical infant’s grip pressure. The softer toys detected approximately 1% of the applied pressure, and the hard toys detected approximately 11% of the applied pressure. Therefore, we can be confident that under more representative conditions, where infants may apply 5 to 35 kPa of grip pressure, our novel implementation of detecting squeezing using a barometric tactile sensor will be able to detect the squeezing of the toys.

## 5. Sequence of Touching

### 5.1. Experimental Setup

A preliminary quantitative experiment was performed using the instrumented toys to detect the sequence of touching. A participant was presented with 4 instrumented toys on a table and asked to touch and play with them (Figure 10). The IMU onboard the P6 sensor embedded in each toy recorded the motion, while a camera simultaneously filmed the toy interaction from an overhead angle to minimise occlusions. The 4 IMUs and the camera were synchronised using the timestamped data from both sensors.

The signals from the IMUs were processed as in Figure 11 to obtain the sequence of touching. First, the signal from the 3-channel accelerometer was combined by calculating the Euclidean norm ${∥a∥}_{2}=\sqrt{a_{x}^{2}+a_{y}^{2}+a_{z}^{2}}$. Then, the DC-offset and noise were filtered using a band-pass filter ($f_{L,1}$ and $f_{H}$), and the signal was passed through a full-wave rectifier to obtain only the positive magnitude of the signal. Finally, the linear envelope was calculated using a low-pass filter ($f_{L,2}$). Similarly, the signal from the 3-channel gyroscope was combined by calculating the Euclidean norm ${∥\omega ∥}_{2}=\sqrt{\omega_{x}^{2}+\omega_{y}^{2}+\omega_{z}^{2}}$, and the linear envelope was calculated using a low-pass filter ($f_{L,3}$).

### 5.2. Results and Discussion

From the video, an independent rater manually coded for the sequence of touches of each toy. These results were our ground truth data for evaluating the performance of the instrumented toys (Table 3). The soft and hard cubes were touched two times, and the soft and hard balls were touched three times.

For the accelerometer, the DC-offset and noise were filtered using $f_{L,1}$ = 0.5 Hz and $f_{H}$ = 10 Hz band-pass filters. The linear envelope for the accelerometer and gyroscope were calculated using $f_{L,2}$ = 0.3 Hz and $f_{L,3}$ = 0.2 Hz low-pass filters, respectively. Threshold values of 0.046 g for the accelerometer and 65 deg/s for the gyroscope were used to identify the initial instance of touching.

From the IMU data (Figure 12), the touching sequence corresponded with the video’s sequence. The timings of the touch from the video were extracted by interpolating from the frame timings, as touch sometimes occurs between frames. In contrast, the timing for the IMU came directly from the data logged with precise discrete timestamping at a high sampling frequency of 64 Hz.

To quantify the accuracy of the IMU results, we computed the timing error and root mean squared error (RMSE) (Table 4). The accelerometer tended to estimate the touch time to be earlier than it was and had an RMSE of 0.32 s, and the gyroscope did a better job at estimating the touch timing with an RMSE of 0.23 s. We could further improve the IMU-based touch detection accuracy by taking the average touch time from the accelerometer and gyroscope, resulting in an RMSE of just 0.19 s.

Therefore, IMU-based touch detection is an accurate way to detect touching and interacting with toys, removing the subjectivity of manual human coders. However, to ensure maximum accuracy and robustness, rather than entirely replacing manually coded data with IMU-based touch detection, the human coders can leverage the IMU data to speed up the manual coding process. In a video recorded at 30 frames per second (fps), the instrumented toys could help narrow the video to a segment spanning approximately 6 to 10 frames to confirm the exact touch time rather than go through the entire video.

## 6. Future Work and Conclusions

In this paper, we presented the initial design and validation of the sensing capabilities of instrumented toys that can be used in lab-based and ecological environments, allowing for a more robust and naturalistic assessment of the development of cognitive flexibility, an aspect of EF, in infants. The results of our study suggested that the instrumented toys were effective at detecting cognitively relevant aspects of infants’ behaviour.

The toy can detect periods of motion to determine when they are touched. From this, the overall sequence of touching can be inferred to calculate MRL and the conditional probability of infant touch sequence to identify mental set-shifting. The toys can also detect when and how much they have been squeezed to further validate infants’ change in mental classification from shape-based to material-based classification.

Such results confirmed the hypothesis that these instrumented toys could enable more quantitative monitoring and measurement of an infant’s EF development. The next step of our iterative design philosophy is to evaluate the instrumented toy’s performance while considering factors such as robustness, reliability, practicality, and ease of use through appropriate studies with mothers and infants in lab-based and home-based environments.

## Figures and Tables

**Figure 1 sensors-23-02709-f001:**
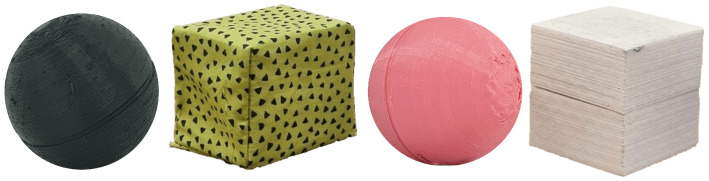
Set of instrumented toy prototypes.

**Figure 2 sensors-23-02709-f002:**
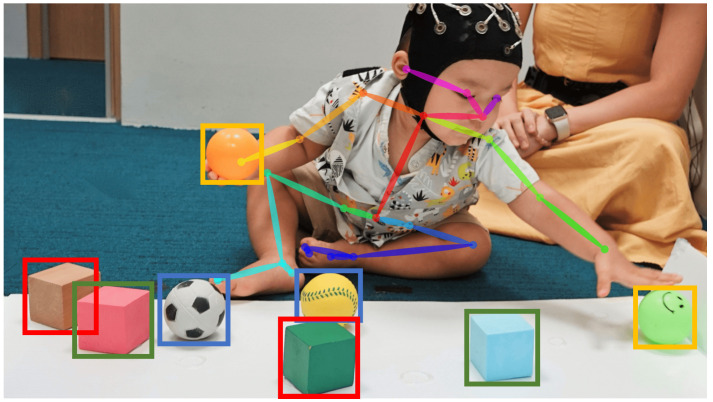
Results of human pose tracking and object detection in object categorisation task videos. (Image used with specific parental consent).

**Figure 3 sensors-23-02709-f003:**
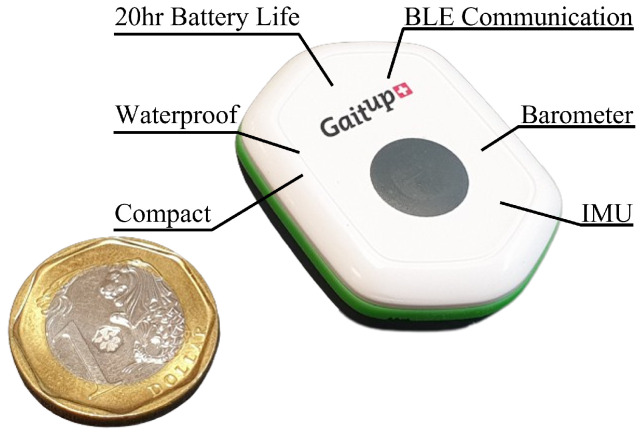
MindMaze Physilog 6^®^ (P6) Sensor.

**Figure 4 sensors-23-02709-f004:**
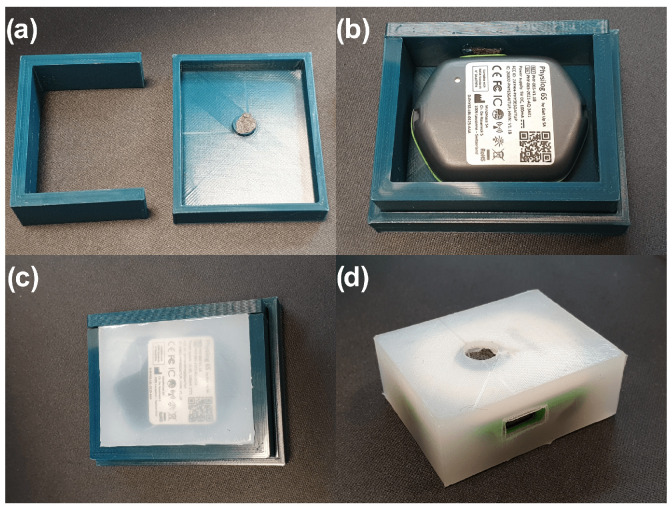
(**a**) 3D Printed mould. (**b**,**c**) P6 sensor enclosed in silicon. (**d**) Fully cured sensor core.

**Figure 5 sensors-23-02709-f005:**
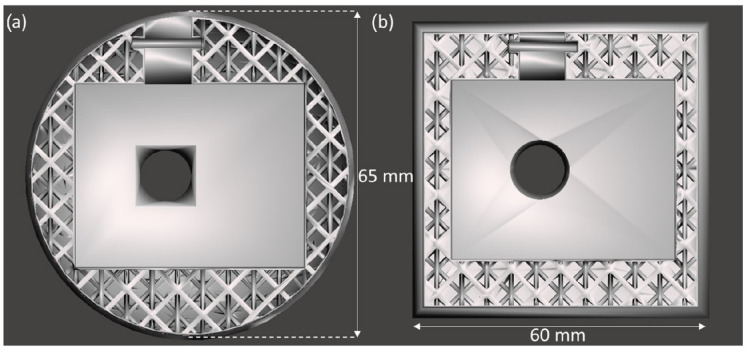
Lattice structure with different outer geometries: (**a**) sphere (**b**) cube.

**Figure 6 sensors-23-02709-f006:**
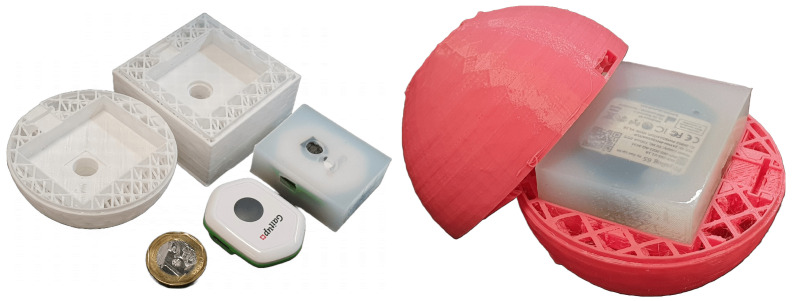
3D printed lattice structure (**left**) and instrumented toy assembly (**right**).

**Figure 7 sensors-23-02709-f007:**
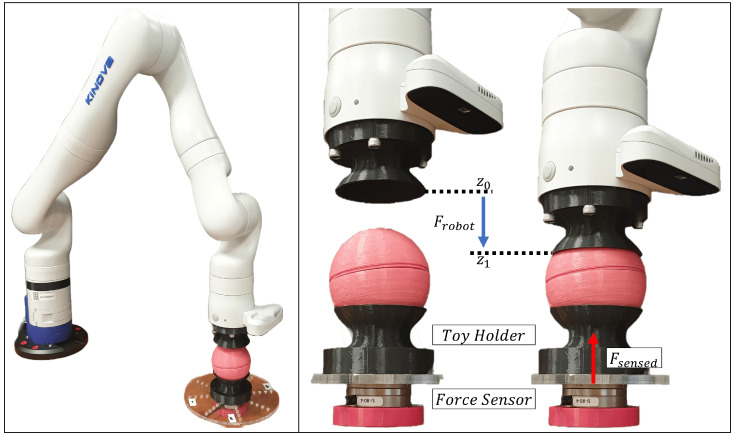
Kinova Gen3 7 DoF Robot applying a squeezing force on instrumented toys.

**Figure 8 sensors-23-02709-f008:**
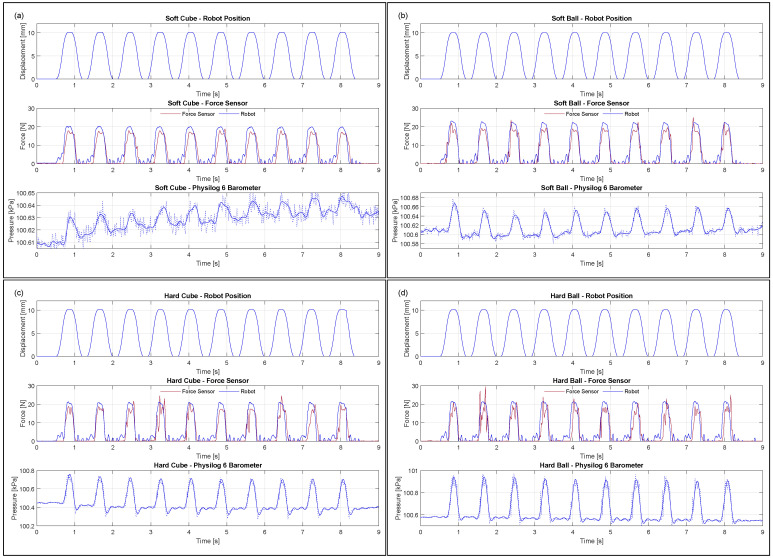
Barometric squeezing detection: (**a**) soft cube, (**b**) soft ball, (**c**) hard cube, (**d**) hard ball.

**Figure 9 sensors-23-02709-f009:**
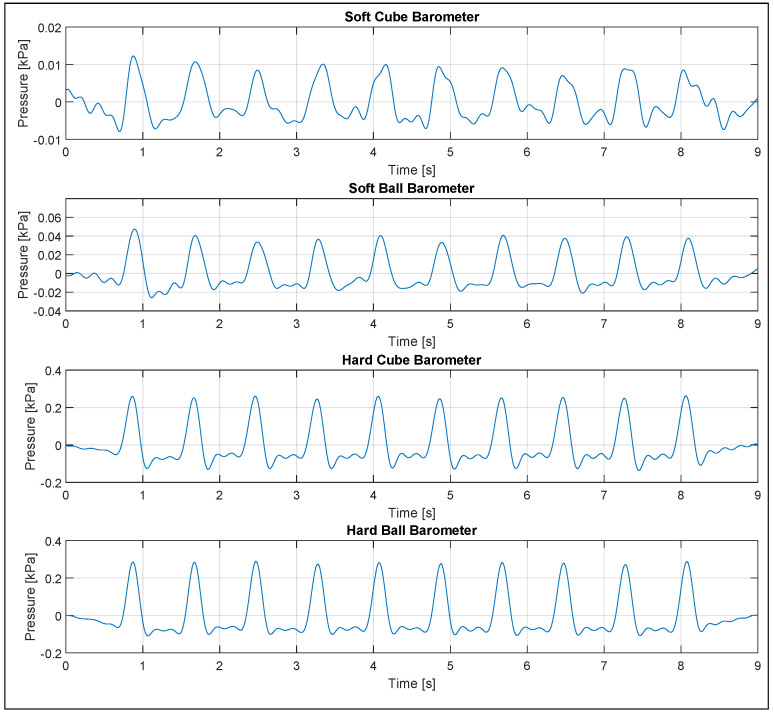
Filtered barometric signals.

**Figure 10 sensors-23-02709-f010:**
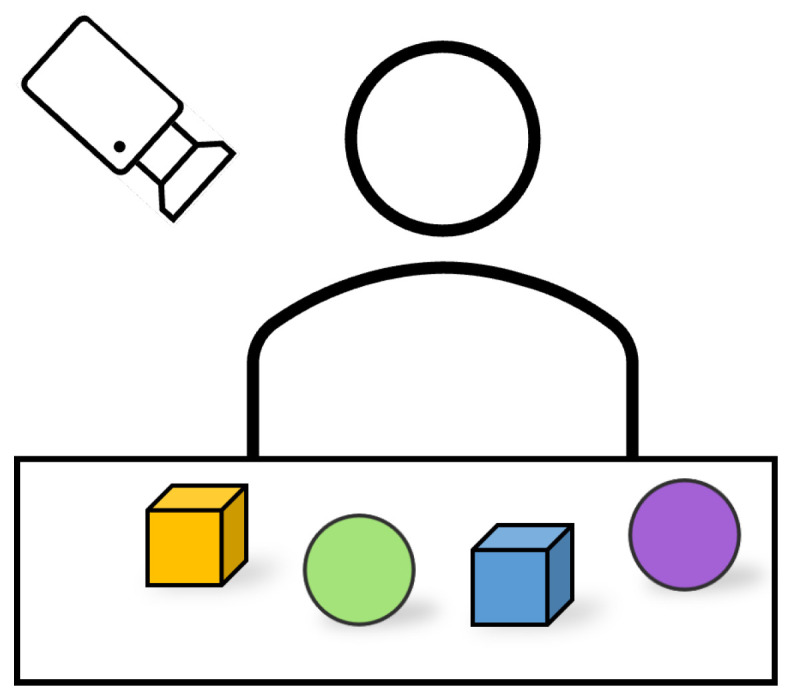
Sequential touching experiment.

**Figure 11 sensors-23-02709-f011:**
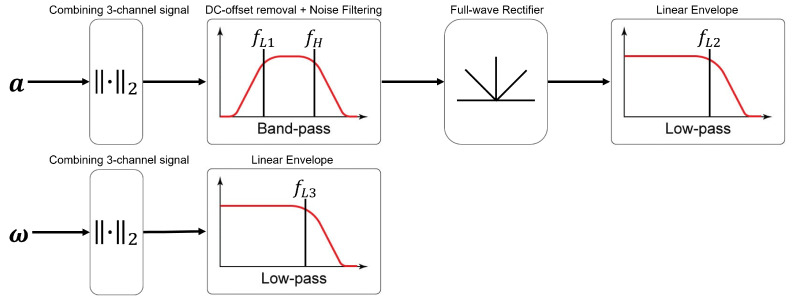
Accelerometer and gyroscope signal processing.

**Figure 12 sensors-23-02709-f012:**
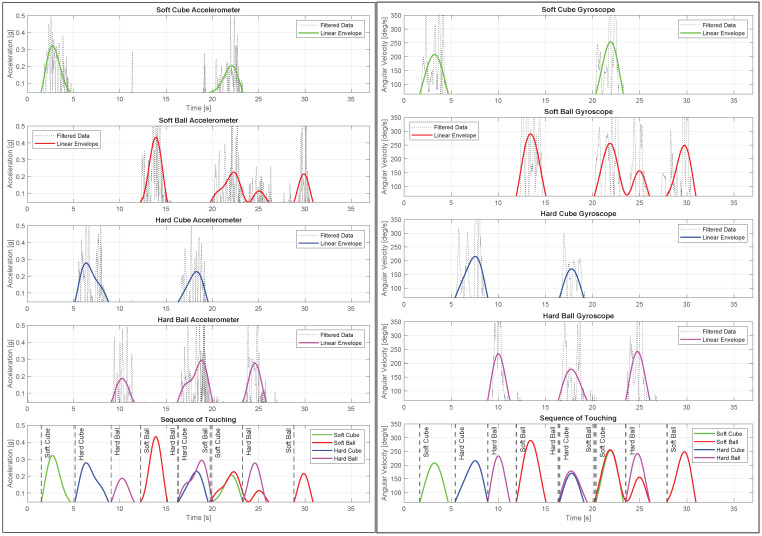
Accelerometer and gyroscope data indicating the sequence of touches.

**Table 1 sensors-23-02709-t001:** Print parameters for the fabrication of lattice structures.

	Nozzle Temperature	Bed Temperature	Print Speed	Layer Thickness
TPU (85A/95A)	$235{\;}^{\circ }$C	$50{\;}^{\circ }$C	30 mm/s	0.2 mm

**Table 2 sensors-23-02709-t002:** Squeezing Results.

Toy	Robot Applied Force (N)	Applied Pressure (Pa)	Measured Force (N)	Barometer Pressure (Pa)	% Pressure Detected
Soft cube.	20.0	2777.78	17.5 (88%)	16.04 ± 2.64	0.58%
Hard cube.	21.0	2916.67	19.0 (90%)	324.60 ± 3.62	11.13%
Soft ball.	22.0	3392.97	18.0 (82%)	51.75 ± 4.53	1.53%
Hard ball.	21.0	3238.74	18.0 (86%)	371.25 ± 5.12	11.46%

**Table 3 sensors-23-02709-t003:** Ground truth touch timing and sequence from video coded by a human rater.

Instrumented Toy	Touch Timing (s)
Soft cube.	1.60
Hard cube.	5.30
Hard ball.	9.30
Soft ball.	12.20
Hard ball.	16.50
Hard cube.	16.60
Soft ball.	20.10
Soft cube.	20.30
Hard ball.	23.70
Soft ball.	28.20

**Table 4 sensors-23-02709-t004:** Touch sequence timing comparison.

Toy	Ground Truth (s)	Accelerometer (s)	Accelerometer Error	Gyroscope (s)	Gyroscope Error	Average (s)	Average Error
Soft cube.	1.60	1.50	−0.10	1.67	0.07	1.59	−0.01
Hard cube.	5.30	5.16	−0.14	5.42	0.12	5.29	−0.01
Hard ball.	9.30	9.06	−0.24	8.87	−0.43	8.97	−0.33
Soft ball.	12.20	12.20	0.00	11.91	−0.29	12.06	−0.14
Hard ball.	16.50	16.25	−0.25	16.35	−0.15	16.3	−0.20
Hard cube.	16.60	16.30	−0.30	16.50	−0.10	16.4	−0.20
Soft ball.	20.10	19.75	−0.35	20.18	0.08	19.97	−0.13
Soft cube.	20.30	19.92	−0.38	20.35	0.05	20.14	−0.16
Hard ball.	23.70	23.30	−0.40	23.50	−0.20	23.4	−0.30
Soft ball.	28.20	28.80	0.60	27.83	−0.37	28.32	0.12
RMSE	—	—	0.32	—	0.23	—	0.19

## Data Availability

The data presented in this study are available on request.

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
