# Peer review of "The Design and Development of Instrumented Toys for the Assessment of Infant Cognitive Flexibility"

_sensors, 2023, doi:10.3390/s23052709_

Round 1

Reviewer 1 Report

This paper presents a set of instrumented toys that can be used in lab-based and ecological environments to study the development of cognitive flexibility. It can complement and accelerate manual means of data coding. The following questions need to be addressed before the article can be considered for publication.

1. The author should consider modifying the structure of the paper. There need to be more explanations about the significance of the research in the introduction. Contents like line 122 can be included in the introduction.

2. There are many pictures of objects in the paper, some of which I think can be combined.

3. The author uses many photographs of objects but needs an intuitive display of each object's name and size. For example, in Figure 4, which objects are sensors? What are the other objects? These things need to be marked out.

4. The authors simulated the sensing effect of toys when infants touched them. Does the author consider the influence of other infant movements on the sensing effect? Such as poking, throwing, etc.

5. How much does the toy cost to make? How long will their production time be? How long does it take to complete an experiment and analysis? It is essential for the practical use of toys.

Reviewer 2 Report

The presented manuscript presents a set of toys that can be used for research in infants. The manuscript is well prepared and very readable, but I have a few comments:

1) only construction models of toys are presented, without tests with the use of infants, moreover, the authors in the title write about Social Interactive Assessment of Infant Cognitive Flexibility but do not present any research in this direction

2) or the presented toys have obtained the required safety certificates

3) in order for the manuscript to be accepted for publication, I propose to change the title and delete Social Interactive Assessment of Infant Cognitive Flexibility.

4) to promote the scientific value of the manuscript, please provide an initial study protocol in infants, preferably based on the applicable pediatric rating scales infant cognitive flexibility

Reviewer 3 Report

The manuscript is academically well-written, and justification of the work is clearly made in the Introduction part. The results were demonstrated in a quite good manner. 

It can be published in present form 

Author Response

The authors thank the reviewer for taking the time to review our paper.

Reviewer 4 Report

This paper presented an investigation of infants' executive function using sensors and commercial equipment. The authors first reviewed the state of the art. Then, the experimental procedures, data processing, and platform were discussed in detail. The authors reported adequate results. However, some clarifications are suggested by the reviewer before publishing this paper. 

1. Could the authors discuss the repeatability of the reported results? For example, did the same kid pick up different objectives when the kid did the experiment for the 2nd time and on a different date?

2. Did the size, color, and distance from the infant impact on the infant's behavior?

Round 2

Reviewer 1 Report

The authors have made the necessary modifications, and the manuscript may be considered for publication.

Reviewer 2 Report

Thank you, I'm satisfied